# Resveratrol Improves the Frozen-Thawed Ram Sperm Quality

**DOI:** 10.3390/ani13243887

**Published:** 2023-12-18

**Authors:** Zhendong Zhu, Haolong Zhao, Haixiang Cui, Adedeji O. Adetunji, Lingjiang Min

**Affiliations:** 1College of Animal Science and Technology, Qingdao Agricultural University, Qingdao 266109, China; zzd2020@qau.edu.cn (Z.Z.); zhl2368447140@163.com (H.Z.); chx1044944467@163.com (H.C.); 2Department of Agriculture, University of Arkansas at Pine Bluff, Pine Bluff, AR 71601, USA; adetunjiadedeji.aa@gmail.com

**Keywords:** resveratrol, antioxidant, cryopreservation, ram, sperm

## Abstract

**Simple Summary:**

Cryopreservation generates a substantial quantity of reactive oxygen species (ROS) in semen, leading to a decline in sperm quality and fertilization capacity. In this study, the influence of resveratrol on thawed ram sperm quality and antioxidant capacity was investigated. The study demonstrated that the supplementation of 50 μM resveratrol significantly enhances the sperm quality and antioxidant capacity of the ram sperm post-thawing. Furthermore, the impact of resveratrol on sirtuin 1 (SIRT1) in thawed sperm was investigated. We found that the activation of adenosine 5‘-monophosphate (AMP)-activated protein kinase (AMPK) by 50 μM resveratrol protected ram sperm from ROS-induced stress. However, at 100 μM, resveratrol appears to reduce sperm motility, mitochondrial membrane potential, acrosomes, and plasma membrane integrity. Consequently, the protective effect of resveratrol on sperm quality or potential cytotoxicity towards ram sperm is contingent on the concentration of resveratrol and filler components, as well as the ability to regulate antioxidant uptake in sperm.

**Abstract:**

Cryopreservation generates a substantial quantity of ROS in semen, leading to a decline in sperm quality and fertilization capacity. The objective of this study was to investigate the effects of resveratrol and its optimal concentration on ram sperm quality after cryopreservation. Ram semen was diluted with a freezing medium containing different concentrations of resveratrol (0, 25, 50, 75, and 100 μM). After thawing, various sperm parameters such as total motility, progressive motility, acrosome integrity, plasma membrane integrity, mitochondrial membrane potential, glutathione (GSH) content, glutathione synthase (GPx) activity, superoxide dismutase (SOD) activity, catalase (CAT) activity, lipid peroxidation (LPO) content, malondialdehyde (MDA) content, ROS level, SIRT1 level, DNA oxidative damage, and AMPK phosphorylation level were assessed. In addition, post-thaw sperm apoptosis was evaluated. Comparatively, the addition of resveratrol up to 75 μM significantly improved the sperm motility and sperm parameters of cryopreserved ram sperm. Specifically, 50 μM resveratrol demonstrated a notable enhancement in acrosome and plasma membrane integrity, antioxidant capacity, mitochondrial membrane potential, adenosine triphosphate (ATP) content, SIRT1 level, and AMPK phosphorylation levels compared to the control group (*p* < 0.05). It also significantly (*p* < 0.05) reduced the oxidative damage to sperm DNA. However, detrimental effects of resveratrol were observed at a concentration of 100 μM resveratrol. In conclusion, the addition of 50 μM resveratrol to the cryopreservation solution is optimal for enhancing the quality of cryopreserved ram sperm.

## 1. Introduction

The success of artificial insemination (AI) crucially depends on the quality of semen preserved in vitro. Cryopreservation technology is pivotal in overcoming temporal and spatial constraints, allowing for the extended preservation of sperm and thereby enhancing the utilization rate of high-quality sheep breeds [1]. This technology facilitates the full exploitation of the reproductive potential of sheep [2]. However, the morphology of sheep’s cervix is a constraint to the use of AI in sheep [3]. Consequently, quality sperm is essential to ensuring successful colonization and migration of sperm through the cervix [4]. One significant challenge during cryopreservation is oxidative stress, primarily induced by elevated levels of ROS, which is the leading cause of sperm damage [5]. It was well known that oxidative stress resulted in a decline in sperm quality and fertilization capacity [6]. The sperm plasma membrane is rich in polyunsaturated fatty acids (PUFA), but low PUFA content in the cytoplasm compromises the antioxidant defense system, making sperm susceptible to lipid peroxidation and subsequent damage [7]. To counteract this, antioxidant compounds are added to cryoprotective solutions [8]. Previous studies have demonstrated that certain oxidants, such as trehalose [9], melatonin [10], and spermine [11], can protect sperm from oxidative stress by enhancing antioxidant capacity. In recent years, polyphenols like tannin extract [12] and tea polyphenol-t [13] have also been shown to exhibit antioxidant properties in sperm.

Among the polyphenols, resveratrol, derived from grape skins and seeds [14], stands out for its antioxidant activity. Resveratrol scavenges free radicals and inhibits lipid peroxidation, thereby safeguarding cellular functionality [15]. Moreover, resveratrol has been demonstrated to enhance the activities of SOD, CAT, GPx, and other antioxidant enzymes within cells [16]. Significant increases in the antioxidant capacity of sperm from humans [17], boars [18], buffaloes [19], and roosters [20] have been observed in cryoprotective fluids containing resveratrol. Research also indicates that resveratrol can protect sperm from oxidative stress by activating AMPK [21]. Furthermore, resveratrol has been shown to improve sperm motility, acrosome integrity, and mitochondrial activity in frozen-preserved giant pandas [22]. In sheep, Al-Mutary et al. [16] demonstrated that resveratrol enhances the quality and fertilization capacity of sheep semen stored at 5 °C. Despite these promising findings, the impact of resveratrol on the cryopreservation of ram semen remains understudied. Therefore, this study is aimed at investigating the effects of resveratrol supplementation in ram semen cryopreservation. It will contribute valuable insights into optimizing the cryopreservation process for ram semen, ultimately enhancing the efficiency of artificial insemination in sheep breeding programs. 

## 2. Materials and Methods

### 2.1. Chemicals 

All chemicals and reagents were purchased from Sigma-Aldrich (Shanghai, China) unless specified otherwise.

### 2.2. Ethical Approval

All animals and experimental procedures were approved by the Qingdao Agriculture University Institutional Animal Care and Use Committee (QAU1121010, 1 October 2019–30 December 2023).

### 2.3. Animals

Eight healthy and fertile rams (small-tailed Han sheep) aged approximately 2 years, utilized for routine artificial insemination in a commercial herd, were used in this study. The rams were kept in an enclosed facility with an evaporative cooling system and separate enclosures. Each ram was fed a commercial diet, and water was freely supplied through tanks.

### 2.4. Collection of Semen

Semen was collected from eight healthy and fertile rams (small-tailed Han sheep) (aged approximately 2 years) two times in one week using an artificial vagina in December of 2022 at the Hongde livestock farm (Shouguang, China). Semen were totally collected 8 times from each ram in this study. A total of 64 ejaculates were obtained and transported to the laboratory in insulated buckets at 37 °C. Sperm motility was analyzed with a computer-assisted sperm analysis (CASA), and only samples with over 80% motility were used in this study. Similarly, a hemocytometer was used to estimate sperm concentration, and only semen with a concentration of more than 2 × 10^9^ sperm/mL was used. The ejaculated semen from the rams was pooled to minimize individual differences, split into 5 parts, and cryopreserved in freezing medium supplemented with different concentrations of resveratrol (0, 25, 50, 75, and 100 μM). The resveratrol was dissolved in dimethyl sulfoxide to create a 200 mM resveratrol solution.

### 2.5. Semen Freezing and Thawing 

The semen samples were diluted in freezing extenders containing 250 mM Tris, 83 mM citric acid, 69 mM fructose, along with 5% (*v*/*v*) glycerol, and 20% (*v*/*v*) egg with varied concentrations of resveratrol (0, 25, 50, 75, and 100 μM) to achieve a sperm concentration of 1 × 10^8^ sperm/mL. Subsequently, the samples were cooled to 4 °C for 3 h and loaded into 0.25-mL straws. Then, the straws were placed horizontally at a height of 5 cm above the surface of liquid nitrogen for 10 min, and then plunged in liquid nitrogen. Thereafter, the straws were stored in a cryogenic storage tank. A week later, the frozen straws were thawed in 37 °C water for 12 s, and after that, the sperm quality was evaluated. 

### 2.6. Evaluation of Sperm Motility

Computer-assisted sperm analysis (CASA) (SCA 20-06-01; Goldcyto, Barcelona, Spain) was performed. For detection, images were acquired using a digital camera (acA780-75gc, Basler, Germany) and a negative phase contrast microscope at 100× magnification, set to a standard parameter of 25 frames/s. The post-thaw sperm was diluted with Tris-citrate-fructose extender before motility evaluation. According to our previous study [23], after preheating the analyzer, semen sample aliquots of 5 µL were added to the Makler chamber. Sperm motility was then assessed in three randomly selected areas using CASA, and more than 500 sperm were evaluated. The percentage of sperm moving at a path speed of 12 µm/s was defined as total sperm motility. Forward movement denotes the percentage of sperm moving in a straight line at a path velocity of 45 µm/s for more than 80% of the time.

### 2.7. Evaluation of Sperm Acrosome Integrity and Plasma Membrane Integrity

According to our previous study [24], sperm acrosome integrity and membrane integrity were detected by fluorescein isothiocyanate-peanut lectin (L-7381, Sigma-Aldrich, Shanghai, China) and the live/dead sperm motility assay kit (L-7011, Thermo Fisher, Shanghai, China), respectively. Briefly, to evaluate sperm acrosome integrity, sperm samples were fixed with methanol solution and incubated with 100 µg/mL fluorescein isothiocyanate-peanut lectin solution and 2.4 mM PI solution for 30 min in the dark before being observed under the microscope. Stained sperm samples were observed and photographed with an epifluorescence microscope (ZEISS DM200LED, Oberkochen, Germany) with 488 nm excitation for FITC-PNA green fluorescence and 535 nm excitation for PI red fluorescence.

For membrane integrity detection, sperm samples were incubated with 100 nM SYBR-14 working solution and 2.4 mM PI solution for 10 min in the dark. The stained sperm were observed and photographed using an epifluorescence microscope (ZEISS DM200LED, Oberkochen, Germany) with 488 nm excitation for SYBR-14 green fluorescence and 535 nm excitation for PI red fluorescence.

### 2.8. Evaluation of Mitochondrial Activity 

The JC-1 mitochondrial membrane potential assay kit (C2003S; Beyotime Institute of Biotechnology; Shanghai, China) was used to analyze sperm mitochondrial activity (ΔΨm) [25]. Additionally, 50 μL of semen was centrifuged at 800× *g* for 5 min, and the supernatant was discarded. The collected sample was washed twice with TCG. Thereafter, 200 μL of JC-1 working solution was added and incubated at 37 °C for 20 min. Sperm was collected by centrifugation at 800× *g* at 4 °C for 5 min and washed twice with JC-1 staining buffer (1×). An appropriate amount of JC-1 staining buffer was used to resuspend the sperm sample. Thereafter, it was placed on ice prior to detection using a flow cytometer. The excitation wavelength was set at 485 nm and the emission wavelength at 590 nm, and a total of 2 × 10^4^ sperm were detected. The experimental data were analyzed using FlowJo-V10, and the whole experimental process was carried out in the dark. All experiments were performed in triplicate (*n* = 3).

### 2.9. Evaluation of ATP Content

The ATP content in sperm was measured using an ATP content assay kit (A095-1-1; Nanjing Jiancheng Bioengineering; Wuhan, China) [23]. Additionally, 30 μL of semen was centrifuged at 800× *g* for 5 min, and the supernatant was discarded while the bottom layer of precipitated cells was collected. Thereafter, 300 μL of cold double-steaming water was added to the cells and placed in ice water to break the homogenate. Then, the cell suspension was heated in boiling water for 10 min, followed by extraction and mixing for 1 min according to the manufacturer’s instructions. Finally, the absorbance was measured at 636 nm with a microplate reader (TECAN, Infinite M Nano, Männedorf, Switzerland). All experiments were performed in triplicate (*n* = 3).

### 2.10. Evaluation of NADH/NAD^+^

According to the NAD(H) assay kit (A114-1-1, Nanjing Jiancheng Bioengineering Institute, Wuhan, China) to analyze sperm NADH/NAD^+^. After thawing 10μL of semen, alkaline extract and homogenate were added and boiled for 5 min. Then, the mixture was cooled in ice water and centrifuged at 10,000× *g* for 10 min at 4 °C, and the supernatant was collected. Equal volumes of acidic extraction solution were added to the supernatant for neutralization and centrifuged at 10,000× *g* for 10 min at 4 °C. Thereafter, the supernatant was collected and placed on ice prior to measurement. Finally, the absorbance was measured at 570 nm with a microplate reader (TECAN, Infinite M Nano, Männedorf, Switzerland). All experiments were performed in triplicate (*n* = 3).

### 2.11. Evaluation of Sperm ROS Content

The ROS content in sperm was measured using an ROS level assay kit (M36008, Thermo Fisher Scientific, Shanghai, China) [26]. Sperm samples were centrifuged and resuspended with 200 μL of working solution. The cells were incubated for 15 min in the dark at 37 °C. Thereafter, the cells were centrifuged and washed three times with 1× PBS. Stained sperm were resuspended in 1× PBS and evaluated by flow cytometry (FACS Aria III, BD Biosciences, San Jose, CA, USA) using a filter with a bandwidth of 574/26 nm, and the measurements denote the mean fluorescence intensity (MFI). All experiments were performed in triplicate (*n* = 3).

### 2.12. Evaluation of Sperm MDA Content

According to our previous study [23], MDA content was measured with a commercial MDA assay kit (S0131S, Beyotime Institute of Biotechnology, Shanghai, China). Briefly, sperm stored at 4 °C were lysed by sonication (20 kHz, 300 W, operating at 50%, 3 min for 10 s on and 5 s off) on ice. The sample was mixed with the preprepared reaction buffer reagent and boiled for 40 min, then centrifuged to collect the supernatant after cooling. The absorbance was measured at 532 nm with a microplate reader (TECAN, Infifinite M Nano, Männedorf, Switzerland). All experiments were performed in triplicate (*n* = 3).

### 2.13. Evaluation of Sperm LPO Content

According to our previous study [27], LPO content was measured with a commercial LPO assay kit (A160-1, Nanjing Jiancheng Bioengineering Institute, Wuhan, China). In brief, normal saline was added to the sperm samples, followed by homogenization in ice water, centrifugation, and mixing with prepared buffer according to the manufacturer’s instructions. The absorbance was measured at 586 nm with a microplate reader (TECAN, Infinite M Nano, Männedorf, Switzerland). All experiments were performed in triplicate (*n* = 3).

### 2.14. Evaluation of GSH Content, GPx, SOD, and CAT Activity

Sperm GSH content, GPx, SOD, and CAT activity were analyzed by the GSH (A006-2-1, Nanjing Jiancheng Bioengineering, China), GPx (A005-1-2, Nanjing Jiancheng Bioengineering, China), SOD (A001-3-2, Nanjing Jiancheng Bioengineering, China), and CAT (A007-1-1, Nanjing Jiancheng Bioengineering, China) assay kit [23]. The sperm samples were centrifuged at 800× *g* for 10 min, and the supernatant was discarded. Thereafter, the collected sperm was washed twice in PBS. Furthermore, precooled RIPA lysate was added to the sperm collected and ground using an electric grinder in ice water, followed by centrifugation at 12,000× *g*. Then, the supernatant was taken for analysis of GSH content, GPx, SOD, and CAT activity according to the manufacturer’s instructions. All experiments were performed in triplicate (*n* = 3).

### 2.15. Evaluation of Sperm Oxidative DNA Damage

Precise quantification of 8-hydroxyguanosine (8-OHdG), a biomarker for oxidative DNA damage, was conducted in sperm samples. The sperm was washed with 1×PBS twice, suspended with 500 μL PBS, and analyzed by flow cytometry after 8-OHdG staining. All experiments were performed in triplicate (*n* = 3).

### 2.16. Western Blotting

Total sperm protein was extracted using sodium dodecyl sulfate (SDS) sample buffer. Moreover, total proteins (20 μg) from each sample were separated on a 10% SDS-PAGE gel (E303-01, Vazyme, Nanjing, China), and the separated proteins were transferred to a polyvinylidene difluoride (PVDF) membrane. Nonspecific binding of PVDF membrane was blocked by TBST containing 5% BSA. Then, 1% BSA (dissolved in TBST) was used to dilute primary antibodies such as anti-SIRT1(13161-1-AP, Proteintech, Wuhan, China) (1:800), anti-AMPK (bs-1115R, Bioss, Beijing, China) (1:800), anti-p-AMPK (AP0432, ABclonal, Wuhan, China) (1:800), anti-p53 (A5761, ABclonal, Wuhan, China) (1:800), caspase 3 (A2156, ABclonal, Wuhan, China) (1:1000), caspase 9 (A0281, ABclonal, Wuhan, China) (1:1000), and incubated for a total of 12 h at 4 °C. Then, the PVDF membranes were placed in a TBST solution for washing. Thereafter, the membranes were incubated with a secondary antibody (AS014, 1:1000, ABclonal, Wuhan, China) for 1 h. ECL plus (ED0016-B, Sparkjade, Jinan, China) was added to the membrane for detection prior to developing the image with a gel imaging analyzer (Alpha, Fluor Chem Q, Shanghai, China).

### 2.17. Statistical Analysis

Data from all replicates were compared using one-way analysis of variance followed by Tukey’s post hoc test (Stat view; Abacus Concepts, Inc., Berkeley, CA, USA). All the values in this study are presented as the mean ± standard error of the mean (SEM). Moreover, treatments were considered to be statistically different from one another at *p* < 0.05.

## 3. Result

### 3.1. Addition of Resveratrol Improved Sperm Motility Parameters 

As shown in Table 1, the motility patterns of sperm analyzed through the movement trajectories generated by CASA showed that the addition of 50 and 75 μM resveratrol to the extender significantly increased (*p* < 0.05) sperm total motility. In addition, the values of progressive motility, curvilinear velocity (VCL), straight-line velocity (VSL), average path velocity (VAP), and wobble (WOB) in the 50 μM resveratrol treatment were higher than those in other treatments. However, values for the progressive motility, VCL, and VSL parameters for other treatments were similar to the control. Moreover, there were no differences (*p* > 0.05) among treatments for the lateral head (ALH), straightness (STR), and linearity (LIN) parameters.

### 3.2. Addition of Resveratrol Improved the Sperm Acrosome Integrity and Plasma Membrane Integrity 

The addition of resveratrol to the extender significantly improved the integrity of the sperm acrosome (*p* < 0.05) after cryopreservation (Figure 1A, Appendix A). Among them, the 50- and 75-μM resveratrol significantly improved sperm acrosome integrity (*p* < 0.05). The addition of resveratrol to the extender up to 75 μM significantly improved (*p* < 0.05) sperm plasma membrane integrity after cryopreservation, with the 75 μM resveratrol treatment having the highest value (Figure 1B, Appendix A).

### 3.3. Addition of Resveratrol Improved Sperm Mitochondrial Activity 

As shown in Figure 2 and Appendix A, the addition of 25, 50, and 75 μM resveratrol to the extender significantly increased (*p* < 0.05) the sperm mitochondrial activity after cryopreservation, and the 75 μM resveratrol treatment presented the highest increase (Figure 2). However, 100 μM resveratrol treatment significantly reduced mitochondrial activity compared to the control group (Figure 2).

### 3.4. Addition of Resveratrol Improved Sperm ATP Content

As shown in Figure 3, it was observed that the addition of resveratrol to the extender significantly improved (*p* < 0.05) the sperm ATP content after cryopreservation. Moreover, the 50 μM resveratrol treatment showed the highest value for ATP content compared to the control (Figure 3).

### 3.5. Addition of Resveratrol Improved Sperm NAD^+^ Content and Reduced Sperm NADH/NAD^+^

As shown in Figure 4A, it was observed that the addition of resveratrol to the extender significantly improved (*p* < 0.05) the sperm NAD+ content after cryopreservation. Moreover, the 75 μM resveratrol treatment showed the highest value of NAD+ content compared to the control (Figure 4A). Additionally, the addition of resveratrol to the extender significantly reduced (*p* < 0.05) the sperm NADH/NAD+ after cryopreservation compared to the control (Figure 4B).

### 3.6. Addition of Resveratrol Reduced Sperm LPO, ROS Level, and MDA Content

As shown in Figure 5A and Appendix A, ROS levels were significantly decreased after the addition of resveratrol up to 75 μM compared with the control group (*p* < 0.05). Notably, the 50 μM resveratrol treatment showed the most significant decrease in ROS levels compared to other treatments. The addition of resveratrol up to 75 μM significantly decreased (*p* < 0.05) the contents of LPO (Figure 5B) and MDA (Figure 5C), with the most significant decrease observed in the 50 μM resveratrol treatment. However, there was no significant difference in LPO content between the 100 μM resveratrol treatment group and the control group (*p* > 0.05). Furthermore, the content of MDA for the 100 μM resveratrol treatment was significantly higher than the control group.

### 3.7. Addition of Resveratrol Improved the Sperm Antioxidative Ability

To investigate the effect of resveratrol on the antioxidant capacity of sperm cryopreservation, the SOD activity, CAT activity, GPx activity, and GSH content were measured. As shown in Figure 6A, the highest SOD activity was observed in the 50 and 75 μM resveratrol treatments (*p* < 0.05). However, the SOD activity for the 25 μM and 100 μM resveratrol treatments and the control were similar. In addition, the addition of resveratrol significantly increased (*p* < 0.05) GPx activity (Figure 6B). The CAT activity in the 50 and 75 μM resveratrol treatment groups significantly increased compared to the other treatments (Figure 6C). Similarly, an increase in the GSH content was also noted with the addition of 50 and 75 μM resveratrol (*p* < 0.05) (Figure 6D).

### 3.8. Addition of Resveratrol Reduced the Oxidative DNA Damage

Analysis of oxidative DNA damage in post-thawed sperm by 8-OHdG staining (Figure 7A–F) showed that the levels of 8-OHdG in post-thawed sperm were significantly different from those in the control (*p* < 0.05) after the addition of 25, 50, and 75 μM resveratrol. The addition of 50 μM resveratrol had a significant difference with other treatment groups (*p* < 0.05). 

### 3.9. Addition of Resveratrol Promotes AMPK Phosphorylation against ROS Damage by Activating SIRT1 

To investigate the mechanism of how resveratrol improves sperm quality, SIRT1 protein expression and AMPK phosphorylation were detected in thawed sperm (Figure 8A–D, Appendix A). Compared with the control group, the SIRT1 expression increased after the addition of 25, 50, and 75 μM resveratrol (*p* < 0.05); however, SIRT1 expression for the 100 μM resveratrol treatment was not different from the control (*p* > 0.05) (Figure 8B). Furthermore, while there was no significant change in total AMPK level with the addition of resveratrol (Figure 8C), the addition of 25, 50, and 75 μM resveratrol significantly increased AMPK phosphorylation after thawing sperm (*p* < 0.05). The highest degree of AMPK phosphorylation was observed in the 50 μM resveratrol treatment group (*p* < 0.05) (Figure 8D). 

### 3.10. Addition of Resveratrol Attenuated the Sperm Apoptosis

Western blotting analysis shows that the expression of apoptosis proteins (caspase-3, caspase-9, and p53) significantly decreased in the resveratrol treatment groups (Figure 9A–D, Appendix A). Among the treatments, the expression of apoptotic proteins was the lowest in the 50 μM resveratrol group for caspase-3 and p53, and the expression of apoptotic proteins was the lowest in the 75 μM resveratrol group for caspase-9.

## 4. Discussion

Artificial insemination in sheep is limited by the morphology of the sheep cervix [28]. The quality of thawed semen is instrumental in determining whether sperm can successfully colonize and migrate within the uterus [29]. The objective of this study was to examine the influence of resveratrol and its optimal concentration on the cryopreservation of ram sperm, focusing on its protective effects against freeze-thaw damage. Results show that the addition of 50 μM resveratrol in a frozen medium significantly improved sperm motility parameters, membrane integrity, acrosome integrity, mitochondrial activity, GPx, CAT, SOD activity, GSH content, SIRT1 expression level, and AMPK phosphorylation level after thawing. Similarly, the levels of LPO, MDA, DNA damage, and apoptosis in sperm decreased.

Sperm motility is important for sperm to swim from the site of ejaculation toward the oviduct for fertilization to take place [30]. In this study, post-thaw sperm motility parameters such as the total motility, progressive motility, VCL, VSL, VAP, and WOB of the 50 μM resveratrol treatment were significantly higher than those of the control. These findings are consistent with Zhu et al. (2019), who reported that the addition of resveratrol improved post-thaw boar sperm motility parameters [31]. In addition, supplementation with 50 μM resveratrol also increased post-thaw ram sperm mitochondrial activity. This result aligns with the findings of Kaeoket et al. (2023), who demonstrated that resveratrol improved sperm mitochondrial bioenergetics [18]. It is well known that the mitochondria are regarded as powerhouses due to their central role in ATP production [32]. It is also notable that sperm motility is dependent on ATP generation [33]. In this study, we found that the post-thaw ram sperm ATP level in the 50-μM resveratrol treatment was higher than that of the control. Thus, we can suggest that the addition of 50 μM resveratrol improved the post-thaw ram sperm motility parameters by maintaining mitochondrial membrane potentials, which are required for ATP generation.

The overproduction of ROS as a result of sperm cryopreservation procedures is a well-documented phenomenon, which significantly increases the risk of oxidative damage and potentially impairs sperm functionality [34]. ROS attack can lead to lipid peroxidation and DNA oxidative damage in sperm, which leads to changes in the structure and function of sperm and ultimately leads to the decline of sperm motility and fertilization ability [35]. Therefore, some studies have explored the addition of antioxidants to diluted semen to reduce ROS accumulation in sperm during cryopreservation to improve sperm motility and fertilization ability [8]. In the present study, the addition of 50 μM resveratrol to the extender protected sperm from oxidative stress by reducing ROS levels and lipid peroxidation. Similar to the present study, Ahmed et al. (2020) showed that the addition of 50–100 μM resveratrol to the extender reduced LPO in sperm [36]. Oxidation [37], heat stress [38], and osmotic stress [39] have been reported to damage the sperm plasma membrane and acrosome during cryopreservation. Nevertheless, the addition of resveratrol to sperm freezing fillers significantly improved the integrity of sperm acrosomes and plasma membranes, which is consistent with the findings of previous studies in humans [17], boar [18], buffalo [19], and rooster [20]. Furthermore, contrary to our findings, Garcez et al. (2010) [40] and Silva et al. (2012) [41] found that resveratrol supplementation did not enhance sperm motility and mitochondrial activity of human and ram sperm after thawing. It has also been documented that 50 μM resveratrol mitigated movement loss but failed to prevent the oxidative damage induced by the cryopreservation of buck sperm [42]. The difference in results may suggest that the protective effect of resveratrol on thawed semen may depend on resveratrol concentration, filler composition, animal species, storage procedures, and the entity under stress conditions.

The present investigation further elucidated that the incorporation of resveratrol into the extender diminished the concentrations of LPO and MDA, concurrently augmenting the GPx, CAT, SOD activity, and GSH content. These results are consistent with studies from other species that suggest that adding resveratrol protects sperm from oxidative damage by boosting their antioxidant capacity [20,31,43]. It is well known that the cellular antioxidant system (scavenging enzymes) is located in the cytoplasm, which is found in sperm in a minimal amount [44]. In addition, cryopreservation decreases the activity of sperm antioxidant enzymes, indicating that sperm had decreased antioxidant capacity and were more susceptible to oxidative damage. Interestingly, our study showed that the addition of resveratrol increased the activity of GPx, CAT, and SOD. These findings are consistent with Zhang et al. (2022) [22], who found that 100 μM of resveratrol significantly improved the activity of GPx and SOD in giant panda sperm.

The metabolism and function of sperm require energy to be maintained during sperm cryopreservation [45]. Resveratrol is an indirect SIRT1 activator [46,47]. The findings of previous research have demonstrated that SIRT1 reduction leads to mitochondrial dysfunction by increasing ROS, LPO, and DNA damage in sperm, resulting in decreased sperm fertilization ability [48]. Similarly, studies by Cuerda et al. have shown that insufficient SIRT1 function in men leads to overactivation of sperm and decreased fertilization ability [49]. Thus, SIRT1 expression was significantly positively correlated with sperm concentration, total motility, and normal sperm morphology. The deficiency of SIRT1 increases the effect of reduced antioxidant defense [50]. In addition, SIRT1 protects sperm from hydrogen peroxide apoptosis through ubiquitination and subsequent degradation of the transcription factor Foxo3a [51]. SIRT1 also inhibits microglia-derived factors through a p53-caspase-3-dependent mechanism, thereby eliminating caspase-mediated apoptosis [52]. Additionally, SIRT1 increases Bcl-2 expression and decreases BAX expression, thereby regulating mitochondrial membrane permeability, mitochondrial function, and cytochrome c release [53]. SIRT1 inhibition caused oocytes to fail to upregulate SOD2 and offset the ROS increase in the presence of increased ROS [54]. Similarly, this study found that adding resveratrol to the freezing additive activated the expression of SIRT1, increased the activity of GPx, CAT, and SOD, reduced the damage of ROS, LPO, and DNA in sperm, and inhibited the expression of P53, caspase-3, and caspase-9. In goat sperm, Price et al. (2012) found that SIRT1 plays a crucial role in activating AMPK and improving mitochondrial function at moderate doses of resveratrol, which is consistent with this study [55]. AMPK is a key kinase that regulates cellular REDOX status by altering metabolic pathways under stress conditions [56]. Additionally, AMPK is documented as a key regulator of sperm physiological function, particularly sperm motility, plasma membrane integrity, and mitochondrial activity [57]. Furthermore, Cantó et al. (2009) indicated that AMPK regulates energy expenditure by modulating NAD^+^ metabolism and SIRT1 activity [58]. NAD^+^ is a metabolic product associated with improved mitochondrial function under somatic stress [59]. Moreover, NAD^+^ levels decline with mitochondrial dysfunction or mitochondrial disease [60]. Activation of AMPK compensates for ATP loss by stimulating catabolic mechanisms and inhibiting synthesis mechanisms [61]. This study also found that adding resveratrol increased ATP content and NAD^+^ levels by activating AMPK. Recently, numerous studies have demonstrated the presence of AMPK in boar [62], ram [45], rooster [63], and mouse [64] sperm. In addition, we showed in a previous study that increased phosphorylation of AMPK in boars improves semen quality [31]. Similarly, in the present study, it was also found that adding resveratrol to the freezing additive improved sperm motility by increasing the phosphorylation level of AMPK. A recent study showed that in boars, the phosphorylated AMPK form is mainly located in the acrosome and equatorial subsegments of the head, as well as in the middle of sperm [65]. However, in roosters, AMPK is present in the acrosome, the middle, and the entire flagella [63]. The distribution of AMPK in different parts of the sperm suggests that, depending on the species, it may promote motility and acrosomal responses. As a result of this variability, the distinct roles of SIRT1 and AMPK in the reproductive activity of different species should be investigated.

## 5. Conclusions

This study showed that 50 μM resveratrol can effectively alleviate the decrease in sperm motility, sperm acrosome integrity, and membrane integrity damage that occurs during cryopreservation of ram semen while maintaining high mitochondrial activity. Additionally, the addition of resveratrol activated AMPK phosphorylation and SIRT1 expression, which is known to decrease ROS production. It also enhanced sperm antioxidant defense systems (e.g., GSH content and activities of GPx, SOD, and CAT) and reduced cell apoptosis and DNA damage.

## Figures and Tables

**Figure 1 animals-13-03887-f001:**
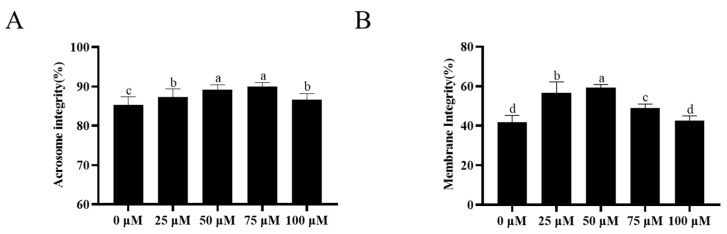
Effect of the addition of different concentrations of resveratrol to extender on acrosome integrity (**A**) and plasma membrane integrity (**B**) of ram sperm after cryopreservation. Values are presented as mean ± standard error of the mean (SEM). Columns with different lowercase letters were significantly different (*p* < 0.05), *n* = 5.

**Figure 2 animals-13-03887-f002:**
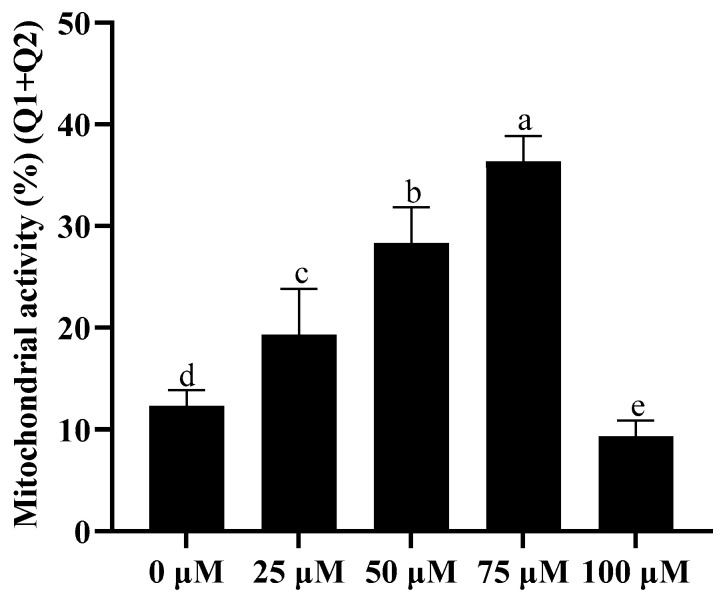
Effect of different concentrations of resveratrol on mitochondrial activity after cryopreservation. Values are presented as mean ± standard error of the mean (SEM). Columns with different lowercase letters are significantly different (*p* < 0.05), *n* = 3.

**Figure 3 animals-13-03887-f003:**
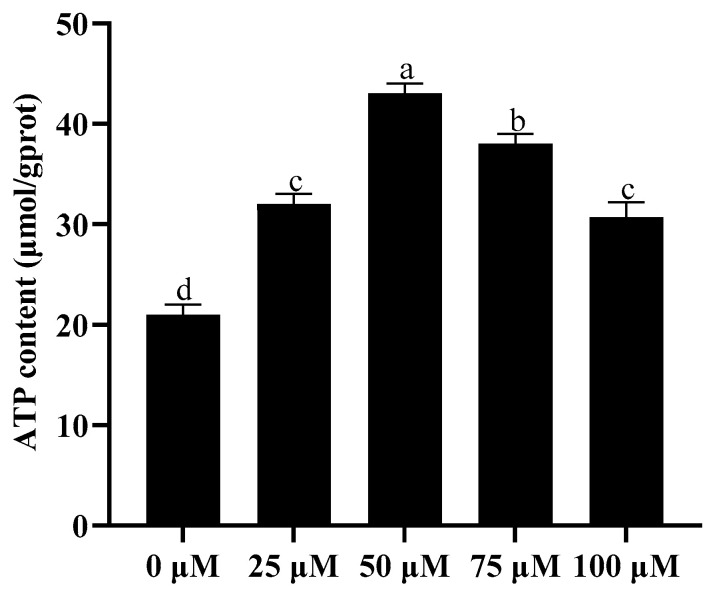
Effect of different concentrations of resveratrol on ATP content after cryopreservation. Values are presented as mean ± standard error of the mean (SEM). Columns with different lowercase letters are significantly different (*p* < 0.05), *n* = 3.

**Figure 4 animals-13-03887-f004:**
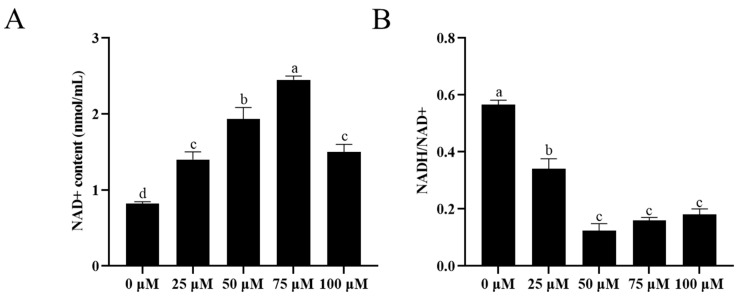
Effect of addition of different concentrations of resveratrol to extender on NAD+ content (**A**) and NADH/NAD+ (**B**) of ram sperm after cryopreservation. Values are presented as mean ± standard error of the mean (SEM). Columns with different lowercase letters were significantly different (*p* < 0.05), *n* = 5.

**Figure 5 animals-13-03887-f005:**
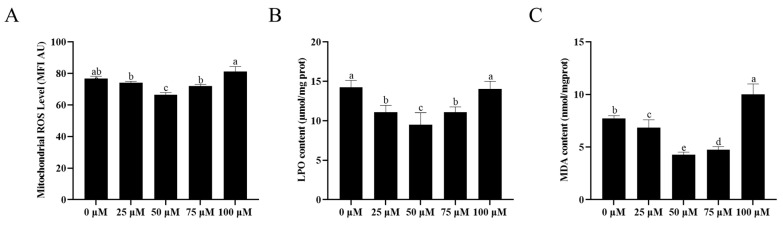
Effect of different concentrations of resveratrol on sperm ROS Level (**A**), LPO content (**B**), and MDA content (**C**) after cryopreservation. Values are presented as mean ± standard error of the mean (SEM). Columns with different lowercase letters were significantly different (*p* < 0.05), *n* = 3.

**Figure 6 animals-13-03887-f006:**
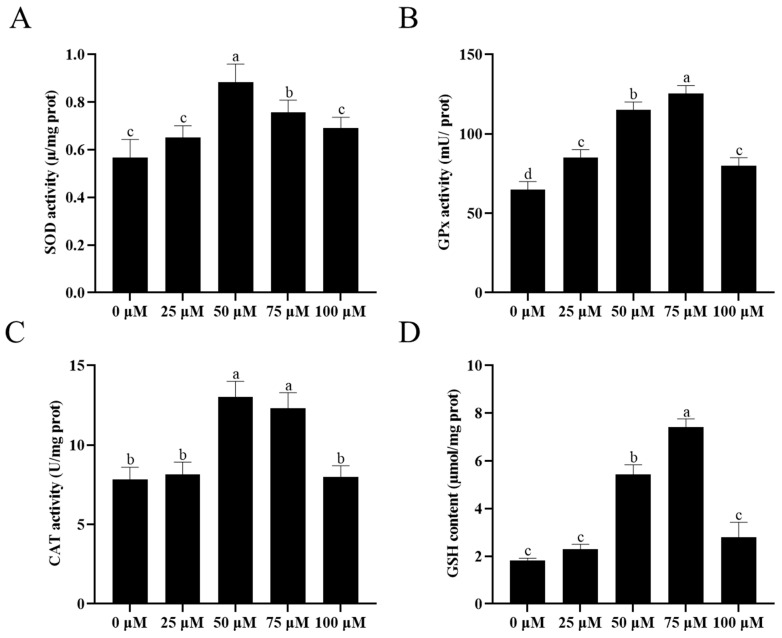
Effect of different concentrations of resveratrol on sperm SOD activity (**A**), GPx activity (**B**), CAT activity (**C**), and GSH content (**D**) after cryopreservation. Values are presented as mean ± standard error of the mean (SEM). Columns with different lowercase letters were significantly different (*p* < 0.05), *n* = 3.

**Figure 7 animals-13-03887-f007:**
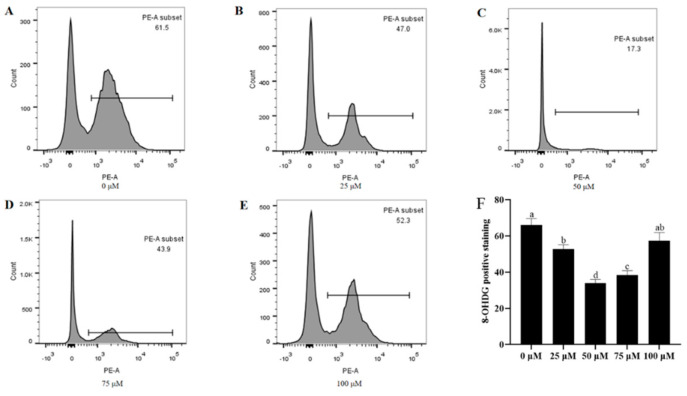
Effect of different concentrations of resveratrol on sperm DNA integrity (**A**–**F**) after cryopreservation. Values are presented as mean ± standard error of the mean (SEM). Columns with different lowercase letters were significantly different (*p* < 0.05), *n* = 3.

**Figure 8 animals-13-03887-f008:**
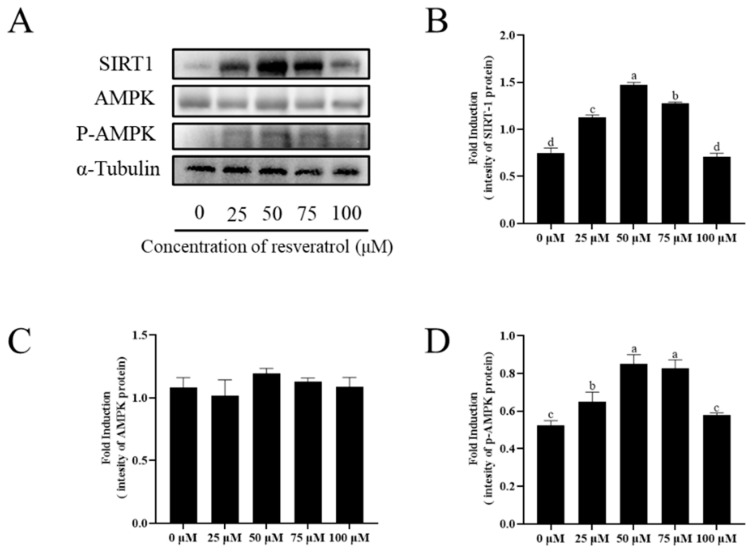
Effect of different concentrations of resveratrol on SIRT1, AMPK, and p-AMPK after cryopreservation. Detection of the expression of SIRT1, AMPK, and p-AMPK by Western blot (**A**–**D**). Columns with different lowercase letters were significantly different (*p* < 0.05), *n* = 3.

**Figure 9 animals-13-03887-f009:**
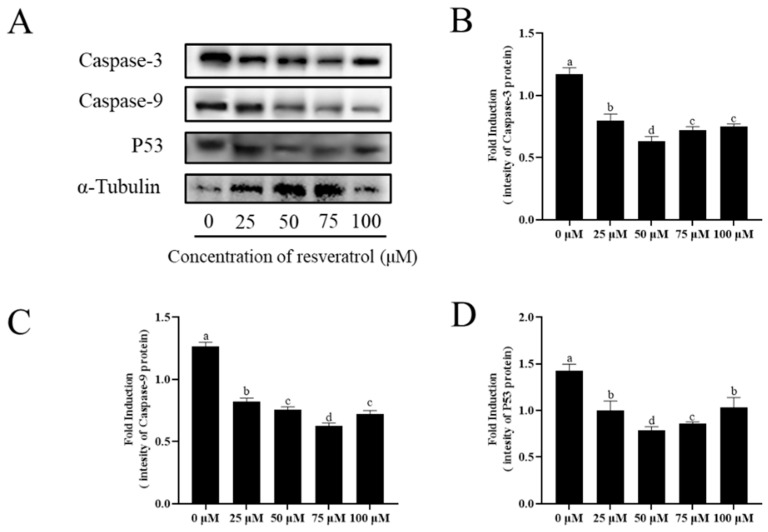
Effect of different concentrations of resveratrol on sperm apoptosis after cryopreservation. Detection of the expression of caspase3, caspase9, and P53 by Western blot (**A**–**D**). Values are presented as mean ± standard error of the mean (SEM). Columns with different lowercase letters were significantly different (*p* < 0.05), *n* = 3.

**Table 1 animals-13-03887-t001:** Effects of different concentrations of resveratrol on post-thaw sperm motility parameters.

	Concentration	0 μM	25 μM	50 μM	75 μM	100 μM
Sperm Parameters						
TM (%)		37.6 ± 1.1 ^b^	40.8 ± 1.8 ^b^	69.2 ± 4.2 ^a^	59.6 ± 5.0 ^a^	33.2 ± 2.4 ^b^
PM (%)		23.6 ± 0.6 ^b^	24.0 ± 3.5 ^b^	48.4 ± 7.0 ^a^	30.4 ± 3.4 ^b^	19.1 ± 0.7 ^c^
VCL (μm/s)		36.7 ± 6.8 ^b^	44.1 ± 13.0 ^b^	61.1 ± 13.1 ^a^	47.1 ± 9.1 ^b^	43.0 ± 5.4 ^c^
VSL (μm/s)		19.3 ± 2.9 ^b^	24.4 ± 7.3 ^b^	34.1 ± 3.6 ^a^	22.3 ± 2.6 ^b^	22.3 ± 5.2 ^b^
VAP (μm/s)		24.2 ± 3.5 ^d^	30.8 ± 9.1 ^b^	35.8 ± 8.8 ^a^	25.2 ± 5.2 ^c^	28.4 ± 5.6 ^c^
BCF (HZ)		7.8 ± 1.7 ^a^	6.4 ± 1.4 ^b^	6.2 ± 1.2 ^b^	6.1 ± 1.1 ^b^	5.4 ± 0.8 ^c^
ALH (μm)		3.0 ± 0.2	3.0 ± 0.4	3.4 ± 0.7	3.8 ± 0.6	3.7 ± 0.2
STR (%)		79.6 ± 0.7	79.1 ± 1.0	77.5 ± 1.3	75.4 ± 1.2	76.6 ± 3.9
LIN (%)		53.7 ± 2.2	56.4 ± 3.9	54.3 ± 3.9	51.1 ± 4.5	51.9 ± 5.3
WOB (%)		67.1 ± 3.1 ^b^	71.1 ± 2.0 ^a^	69.8 ± 3.8 ^a^	67.6 ± 5.5 ^b^	67.2 ± 4.1 ^b^

Values are expressed as mean ± standard error. Different letters within the same row indicate significant differences (*p* < 0.05). VCL, curvilinear velocity; VSL, straight-line velocity; VAP, average path velocity; BCF, beat-cross frequency; ALH, lateral head; STR, straightness (VSL/VAP); LIN, linearity (VSL/VCL); WOB, wobble (VAP/VCL).

## Data Availability

Data are contained within the article and Appendix A.

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
