# Peer review of "Resveratrol Improves the Frozen-Thawed Ram Sperm Quality"

_animals, 2023, doi:10.3390/ani13243887_

Round 1

Reviewer 1 Report

Comments and Suggestions for Authors

Study named Resveratrol improves the frozen-thawed sheep Sperm quality was conducted to examine the influence of different concentration of antioxidant resveratrol on cryopreservation of ram spermatozoa focusing on its protective effects against freeze/thaw damage. The manuscript covers a subject matter of interest, authors did put great effort on performing large amount of analysis that generated large amount of data. However, it needs to be expand more on materials and methods before it should be considered for publication.

Experimental design is not clear. The author stated 8 rams and 15 replicates. What was the frequency of collection? Were there 2 repetitions or all rams were collected on the same day, once or twice (8x2=16 and there were 15 samples) and all samples were pooled and tested on the same day?

When was the experiment conducted? As sheep is seasonal it is important to know whether experiment is conducted in breeding or in non-breeding season.

Also, the number of animals could be a problem of this study. Did the authors performed power test to determine minimum number of animals/samples?

What is pseudovaginal method? Is it artificial vagina?

Collected semen was selected based on concentration and motility, authors should indicate the methods and equipment used.

All 15 samples were pooled and divided in 5 aliquots that was diluted with basic medium with different resveratrol concentration? This also should be clarified. Samples were not prediluted and centrifuged. Raw semen was pooled and diluted with extender?

After holding the straws on rack above nitrogen and before storing it in tank it should be plunged in liquid nitrogen, please add this information.  

Regarding sperm motility, samples were not diluted prior to analysis? What is maker chamber? Do you mean Makler? Which parameters were analyzed? In discussion there are no information on this results and interaction with other parameters like mitochondrial activity etc.

In conclusions you stated that resveratrol prevents the damage of apical body membrane. You don’t mention previously in text apical body membrane. What do you mean by that, acrosome membrane?

In discussion (L310) you stated that 50 µL of resveratrol showed protective effect while in conclusions there were both 50 and 75.

Instead of sheep sperm it is advised to use ram sperm. 

Author Response

Response to the Reviewer 1’s comments

Point 1: Experimental design is not clear. The author stated 8 rams and 15 replicates. What was the frequency of collection? Were there 2 repetitions or all rams were collected on the same day, once or twice (8x2=16 and there were 15 samples) and all samples were pooled and tested on the same day?

[Response 1] Sorry for the confusion. We have elaborated on the collection of semen. Please see L90-100.

“Semen was collected from eight healthy and fertile rams (Small-Tailed Han sheep) (aged about 2 years) twice a week using an artificial vagina during the breeding season at the Hongde livestock farm (Shouguang, China). A total of 16 ejaculates were obtained, with each ram's ejaculate kept separately and incubated at 37 °C during transport to the laboratory. Computer-assisted sperm analysis (CASA) was used to assess ejaculated semen motility, and only samples with over 80% motility were used. With the aid of an hemocytometer for sperm concentration detection, only semen with a density exceeding 2 x 109 sperm/mL were used. The ejaculated semen from eight rams were pooled to minimize individual differences, split into 5 parts and cryo-preserved in freezing medium supplemented with different concentrations of resveratrol (0, 25, 50, 75 and 100 μM). The resveratrol was dissolved in dimethyl sulfoxide to make 200 mM resveratrol solution”.

Point 2: When was the experiment conducted? As sheep is seasonal it is important to know whether experiment is conducted in breeding or in non-breeding season.

[Response 2]Thank you very much for your comments. The experiment was conducted in the ram breeding season (2022.09 to 2023.02), We added the information in the revision. Please see L90-92.

“Semen was collected from eight healthy and fertile rams (Small-Tailed Han sheep) (aged about 2 years) twice a week using an artificial vagina during the breeding season at the Hongde livestock farm (Shouguang, China)”.

Point 3: Also, the number of animals could be a problem of this study. Did the authors performed power test to determine minimum number of animals/samples?

[Response 3]:Thank you very much for your comments. We used eight rams for semen collection in this study.  In fact, during the pre-experiment, we collected semen on three different occassions from 35 rams to evaluate the rams raw semen quality at the Hongde livestock farm. Thereafter, we selected eight rams for the experiment based on their good raw semen quality (sperm motility, ejaculated volume, sperm concentration) and age factor. Because our study is foucused on the effect of resveratrol on ram sperm cryopreservation, the raw semen quality is very important. More so, to prevent bias or individual differences we mixed/pooled the semen as described in the methods. Kindly note that the number of animals and the methodology adopted has been published in high reputable journals.

  1. Effect of different concentrations of resveratrol on the quality and in vitro fertilizing ability of ram semen stored at 5 °C for up to 168 h” https://www.sciencedirect.com/science/article/pii/S0093691X2030279X
  2. “Effect of antioxidants resveratrol and quercetin on in vitro evaluation of frozen ram sperm" https://pubmed.ncbi.nlm.nih.gov/22289215/
  3. “Effects of resveratrol on DLD and NDUFB9 decrease in frozen semen of Mongolian sheep” used 3, 4, and 3 rams respectively for their experiments. https://pubmed.ncbi.nlm.nih.gov/37956782/

Point 4: What is pseudoganglia method? Is it artificial vagina?

[Response 4]:Yes, it is the artificial vagina, we revised the sentence. Please see L91. “Semen was collected from eight healthy and fertile rams (Small-Tailed Han sheep) (aged about 2 years) twice a week using an artificial vagina during the breeding season at the Hongde livestock farm (Shouguang, China).”

Point 5: Collected semen was selected based on concentration and motility, authors should indicate the methods and equipment used.

[Response 5]:Thank you very much for your suggestion. We added the information in the revision accroding to your suggestion. Please see L92-97.

 “A total of 16 ejaculates were obtained, each ram's ejaculate was kept separately and incubated at 37 °C during transport to the laboratory. Computer-assisted sperm analysis (CASA) was used to assess ejaculated semen motility, and only samples with over 80% motility were used. With the aid of an hemocytometer for sperm concentration detection, only semen with a density exceeding 2 x 109 sperm/mL were used.

Point 6: All 15 samples were pooled and divided in 5 aliquots that was diluted with basic medium with different resveratrol concentration? This also should be clarified. Samples were not prediluted and centrifuged. Raw semen was pooled and diluted with extender?

[Response 6]: Thanks for your suggestion. The raw semen from eight rams was pooled to minimize individual differences, split into 5 parts and cryo-preserved in freezing medium supplemented with different concentrations of resveratrol (0,25, 50, 75 and 100 μM) in this study. The raw semen samples were not prediluted and centrifuged.

We added the information in the revision, please L97-100.

   “The ejaculated semen from eight rams was pooled to minimize individual differences, split into 5 parts and cryo-preserved in freezing medium supplemented with different concentrations of resveratrol (0, 25, 50, 75 and 100 μM). The resveratrol was dissolved in dimethyl sulfoxide to make 200 mM resveratrol solution”

Point 7: After holding the straws on rack above nitrogen and before storing it in tank it should be plunged in liquid nitrogen, please add this information. 

[Response 7]:Thank you very much. We added the information in the revised manuscript according to your suggestion. Please see L107-109.

“Subsequently, the straws were horizontally at a height of 5 cm above the surface of liquid nitrogen for 10 min, and then plunged in liquid nitrogen. After that, the straws were stored in the cryogenic storage tank.”

Point 8: Regarding sperm motility, samples were not diluted prior to analysis? What is maker chamber? Do you mean Makler? Which parameters were analyzed? In discussion there are no information on this results and interaction with other parameters like mitochondrial activity etc.

[Response 8]:Thanks for your comments and suggestions. The post-thaw sperm was diluted with Tris-citrate-fructose extender before motility evaluation. Yes, it is Makler counting chamber. We analyzed the following parameters; Sperm total motility, progressive motility, curvilinear velocity (VCL), straight-line velocity (VSL), average path velocity (VAP), beat-cross frequency (BCF), lateral head (ALH), straightness (STR), linearity (LIN) and wobble (WOB) in this study. Moreover, we have revised the manuscript accordingly. In addition, we added information about the motility parameters and interaction with mitochondrial membrane potentials and ATP levels in the discussion section, please see the revision. L370-384.

“Sperm motility is important for sperm to swim from the site of ejaculation toward the oviduct for the fertilization to take place[30]. In this study, post-thaw sperm motility parameters such as the total motility, progressive motility, VCL, VSL, VAP, and WOB of the 50 μM resveratrol treatment were significantly higher than those of control. These findings are consistent with Zhu et al. (2019), who reported that the addition of resveratrol improved pos-thaw boar sperm motility parameters[31]. In addition, supplementation of 50 μM resveratrol also increased post-thaw ram sperm mitochondrial activity. This result aligns with the findings of Kaeoket et al. (2023),who demonstrated that the resveratrol improved the sperm mitochondrial bioenergetics[18]. It is well known that the mitochondria are regarded as powerhouses due to their central role in ATP production [32]. It is also notable that sperm motility is dependent on ATP generation[33]. In this study, we found that post-thaw ram sperm ATP level in 50 μM resveratrol treatment was higher than that of control. Thus, we can suggest that the addition of 50 μM resveratrol improved the post-thaw ram sperm motility parameters by maintaining mitochondrial membrane potentials which is required for ATP generation”.

Point 9: In conclusions you stated that resveratrol prevents the damage of apical body membrane. You don’t mention previously in text apical body membrane. What do you mean by that, acrosome membrane?

[Response 9]:Sorry for the confusion. We revised the sentence, please see revision L 465. “sperm acrosome integrity and membrane integrity damage”

Point 10: In discussion (L310) you stated that 50 µL of resveratrol showed protective effect while in conclusions there were both 50 and 75.

[Response 10]:Sorry for the confusion. It was 50 μM resveratrol that showed the protective effect we referred to. We have revised the conclusion. Please see the revision. L464-470.

“This study showed that 50 μM resveratrol can effectively alleviate the decrease in sperm motility and sperm acrosome integrity and membrane integrity damage that occur during cryopreservation of ram semen while maintaining a high mitochondrial activity. Also, the addition of resveratrol activated AMPK phosphorylation and SIRT1 expression, which is known to decrease ROS production. It also enhanced sperm antioxidant defense systems (e.g., GSH content and activities of GPx, SOD, and CAT) and reduced cell apoptosis and DNA damage.

Point 11: Instead of sheep sperm it is advised to use ram sperm. 

[Response 11]:Thank you very much for your suggestion. We revised the manuscript accordingly. Please see the revision.

Reviewer 2 Report

Comments and Suggestions for Authors

The authors of the manuscript intended to investigate the effects of resveratrol on sheep sperm quality after cryopreservation identifying the optimal concentrations to be added to the freezing medium. The supplementation of 0, 25, 50, 75, and 100 μM concentrations of resveratrol were tested. The authors concluded that 50 and 75 μM resveratrol concentrations can effectively alleviate the decrease of sperm motility and the damage of ram spermatozoa membranes during cryopreservation maintaining high mitochondrial activity. Also, the addition of resveratrol has activated the expression of AMPK phosphorylation and SIRT1, which has been associated to the decrease of ROS production. The sperm antioxidant defense systems (e.g., GSH content and activities of GPx, SOD, and CAT) was also enhanced, and cell apoptosis and DNA damage reduced.

This manuscript has an interesting and up to date topic, although not very original. Also, the topic falls within the scope of the Animals with a suitable methodology. However, the experimental design presents some flaws. Therefore, I recommend this manuscript for publication after a revision.

Specific recommendations 

L90 What do you mean by pseudovaginal method? Briefly explain this method.

L91 Please specify the year station and location of the collection was performed.

L84-92 Why did the authors pooled the ejaculates? Did they put together all the ejaculates obtained from the 8 rams? Did they get 2 ejaculates from each ram? So, why they only had 15 ejaculates? Please explain this methodology in detail.

Moreover, if the ejaculates collected from each animal were individually used in the experimental design, it would be possible to know the effect of the treatment and of the animal, as well as the interaction between the two.

L99 How did the authors performed the dilution of resveratrol? Did they use an excipient? Please explain in detail

L221 According to the authors, some data were obtained from 5 replicates as referred in figure 4 (NAD+ content). Please correct.

Table and figures. To a better understanding of results, the table and figures should be placed in the results section specifically near the text where they were described. Is it possible to reformulate to a more reader-friendly version?

Comments on the Quality of English Language

Minor englhish editing should be performed

Author Response

Response to the Reviewer 2’s comments

Point 1: L90 What do you mean by pseudovaginal method? Briefly explain this method.

[Response 1]:Sorry for the miscommunication. it is the artificial vagina, we revised the sentence. Please see L91. “Semen was collected from eight healthy and fertile rams (Small-Tailed Han sheep) (aged about 2 years) twice a week using an artificial vagina during the breeding season at the Hongde livestock farm (Shouguang, China).”

Point 2: L91 Please specify the year station and location of the collection was performed.

[Response 2]:Thank you very much for pointing this out. We added the information in the revision following with your suggestion. Please see the revision. L90-92.

“Semen was collected from eight healthy and fertile rams (Small-Tailed Han sheep) (aged about 2 years) twice a week using an artificial vagina during the breeding season at the Hongde livestock farm (Shouguang, China)”.

Point 3: L84-92 Why did the authors pooled the ejaculates? Did they put together all the ejaculates obtained from the 8 rams? Did they get 2 ejaculates from each ram? So, why they only had 15 ejaculates? Please explain this methodology in detail.

[Response 3]:Thank you very much for your suggestion. We are sorry for the confusion. In this study, 8 healthy and fertile rams (Small-Tailed Han rams) aged about 2 years were used for semen collection during the breeding season at the Hongde livestock farm (Shouguang, China). 2 ejaculates were collected from each ram (8x2=16 ejaculates). The samples were tested for sperm motility and sperm concentration, and only sperm with motility over 80% and concentration over 2 x 109 sperm/mL were mixed and used for the experiment. We added the information in the revision. Please see L90-99.

“Semen was collected from eight healthy and fertile rams (Small-Tailed Han sheep) (aged about 2 years) twice a week using an artificial vagina during the breeding season at the Hongde livestock farm (Shouguang, China). A total of 16 ejaculates were obtained, with each ram's ejaculate kept separately and incubated at 37 °C during transport to the laboratory. Computer-assisted sperm analysis (CASA) was used to assess ejaculated semen motility, and only samples with over 80% motility were used. With the aid of an hemocytometer for sperm concentration detection, only semen with a density exceeding 2 x 109 sperm/mL were used. The ejaculated semen from eight rams were pooled to minimize individual differences, split into 5 parts and cryo-preserved in freezing medium supplemented with different concentrations of resveratrol (0, 25, 50, 75 and 100 μM). The resveratrol was dissolved in dimethyl sulfoxide to make 200 mM resveratrol solution”.

Point 4: Moreover, if the ejaculates collected from each animal were individually used in the experimental design, it would be possible to know the effect of the treatment and of the animal, as well as the interaction between the two.

[Response 4]:Thank you very much for your comments and suggestions. In present study, we focused on whether and how the resveratrol affected the ram sperm quality during cryopreservation, thus, to minimize individual differences, we used the mixed semen to do the experiment. We appreciate your thought, but unfortunately, in this present study we pooled the semen. It is our desire to use individual ram ejaculates for our future sperm study to understand animal differences based on your suggestion. Thank you again. 

Point 5: L99 How did the authors performed the dilution of resveratrol? Did they use an excipient? Please explain in detail

[Response 5]:Thank you very much for your suggestion. We added the information in the revision, please see L99-100.

“The resveratrol was dissolved in dimethyl sulfoxide to make 200 mM resveratrol solution.”

Point 6: L221 According to the authors, some data were obtained from 5 replicates as referred in figure 4 (NAD+ content). Please correct.

[Response 6]:Thank you very much for your suggestion. We revised it. Please see the revision. L227

Point 7: Table and figures. To a better understanding of results, the table and figures should be placed in the results section specifically near the text where they were described. Is it possible to reformulate to a more reader-friendly version?

[Response 7]:We sincerely thank you for your suggestion. We placed tables and figures below the corresponding results section in the revision. Please see the revision. Thank you again.

Round 2

Reviewer 1 Report

Comments and Suggestions for Authors

The authors did improve the manuscript, but there are some things that needs to be revised. 

L91 authors stated that rams were collected twice a week during breeding season. This sounds as they were collected repeatedly during consecutive weeks and not just two times in one week. This should be rephrased. 

Also, sections 2.4 and 2.5 should be rewritten, maybe with the aid of English speaker familiar with scientific writing. This paragraphs are now more clear, but some phrases sounds better in old version (for example: samples were loaded in straws instead of placed in straws , cooled for 2 hours instead of for duration of 3 hours.). Phrases like thinned (diluted) should be avoided.  

Author Response

Response to the Reviewer1’s comments

  1. L91 authors stated that rams were collected twice a week during breeding season. This sounds as they were collected repeatedly during consecutive weeks and not just two times in one week. This should be rephrased.

[Response 1]:Thank you very much for your suggestion. We revised the sentence and added the detail information in the revision. In this study, we collected semen from eight rams on December of 2022. The frequency of collect semen from each ram was twice per week, thus totally 64 ejaculates were used in this study. Please see the revision. L90-94

“Semen was collected from eight healthy and fertile rams (Small-Tailed Han sheep) (aged about 2 years) two times in one week using an artificial vagina on December of 2022 at the Hongde livestock farm (Shouguang, China). Semen were totally collected 8 times from each ram in this study. A total of 64 ejaculates were obtained and transported to the laboratory in insulated buckets at 37°C”

  1. Also, sections 2.4 and 2.5 should be rewritten, maybe with the aid of English speaker familiar with scientific writing. This paragraphs are now more clear, but some phrases sounds better in old version (for example: samples were loaded in straws instead of placed in straws, cooled for 2 hours instead of for duration of 3 hours.). Phrases like thinned (diluted) should be avoided.

[Response 2]:Thank you very much for your suggestion. We revised the 2.4 and 2.5 sections. And Dr. Adedeji O. Adetunji, an assistant professor at University of Arkansas at Pine Bluff, helps to edited these two sections. Please see the revision. Please see L89-110.

“2.4. Collection of semen

Semen was collected from eight healthy and fertile rams (Small-Tailed Han sheep) (aged about 2 years) two times in one week using an artificial vagina on December of 2022 at the Hongde livestock farm (Shouguang, China). Semen were totally collected 8 times from each rams in this study. A total of 64 ejaculates were obtained and transported to the laboratory in insulated buckets at 37°C. Sperm motility was analyzed with a computer-assisted sperm analysis (CASA), and only samples with over 80% motility were used in this study. Similarly, a hemocytometer was used to estimate sperm concentration and only semen with a concentration more than 2 × 109 sperm/mL was used. The ejaculated semen from the rams were pooled to minimize individual differences, split into 5 parts and cryo-preserved in freezing medium supplemented with different concentrations of resveratrol (0, 25, 50, 75 and 100 μM). The resveratrol was dissolved in dimethyl sulfoxide to make 200 mM resveratrol solution.”

“2.5. Semen freezing and thawing

The semen samples were diluted in freezing extenders containing 250 mM Tris, 83 mM citric acid, 69 mM fructose, along with 5% (v/v) glycerol and 20% (v/v) egg with varied concentrations of resveratrol (0, 25, 50, 75, and 100 μM) to achieve a sperm con-centration of 1 × 108 sperm/mL. Subsequently, the samples were cooled to 4 °C for 3 h and loaded in 0.25 mL straws. Then, the straws were placed horizontally at a height of 5 cm above the surface of liquid nitrogen for 10 min, and then plunged in liquid nitrogen. Thereafter, the straws were stored in a cryogenic storage tank. A week later, the frozen straws were thawed at 37 °C water for 12 s, after that the sperm quality was evaluated.”

Reviewer 2 Report

Comments and Suggestions for Authors

The authors have attended to most of my suggestions. Therefore the article should be published

Comments on the Quality of English Language

Minor editing of English language should be performed

Author Response

 Minor editing of English language should be performed

[Response 1]:Thank you very much for your suggestion. We edited the English writing with the help of Dr. Adedeji O. Adetunji, who an assistant professor at University of Arkansas at Pine Bluff. Please see the revision. Thank you again.